# DISCRET: A SELF-INTERPRETABLE FRAMEWORK FOR TREATMENT EFFECT ESTIMATION

## ABSTRACT

Individual treatment effect is of great importance for healthcare and beyond. While most existing solutions focus on accurate treatment effect estimations, they rely on non-interpretable black-box models that can hinder stakeholders from understanding the underlying factors driving the prediction. To address this issue, we propose DISCRET, a self-interpretable framework that is inspired by how stakeholders make critical decisions in practice. DISCRET identifies samples similar to a target sample from a database by using interpretable rules and employs their treatment effect as the estimated ITE for the target sample. We present a deep reinforcement learning-based rule learning algorithm in DISCRET to achieve accurate ITE estimation. We conduct extensive experiments over tabular, natural language, and image settings. Our evaluation shows that DISCRET not only achieves comparable performance as black-box models but also generates more faithful explanations than state-of-the-art post-hoc methods and self-interpretable models.

## 1 INTRODUCTION

Individual treatment effect (ITE) quantifies the difference between one individual's outcome with and without receiving treatment. Estimating ITE is a significant problem not only in healthcare but also in other domains (Basu et al., 2011; Pryzant et al., 2021; Feder et al., 2021; Jerzak et al., 2023b;a). It serves as an important tool for tailoring interventions to the specific needs of each individual. A substantial body of literature has been dedicated to investigating various methodologies for accurately estimating ITE by harnessing the potential of machine learning and deep learning techniques (Shalit et al., 2017; Yoon et al., 2018; Zhang et al., 2022; Liu et al., 2022).

However, the estimation of ITE confronts a significant challenge with respect to interpretability and explainability. Due to the black-box nature, machine learning models hinder stakeholders from understanding the underlying factors and mechanisms driving treatment effect predictions. Efforts to develop interpretable machine learning methods and frameworks are therefore essential to ensure that ITE estimations are not only accurate but also trustworthy (Kim & Bastani, 2019; Crabbé et al., 2022; Chen et al., 2022).

The state-of-the-art interpretable ITE estimation approaches, while making significant strides in enhancing explainability, often struggle to provide highly accurate ITE estimations (Athey & Imbens, 2016; Nie & Wager, 2021). Additionally, recent studies have made concerted efforts to elucidate the inner workings of black-box models in estimating ITE through post-hoc explanations (Kim & Bastani, 2019). However, post-hoc explanations have also been blamed for their potential lack of faithfulness (Rudin, 2019; Bhalla et al., 2023), which is also empirically validated in Section 4.

To address the above issues, we propose a self-interpretable ITE estimation framework, DISCRET, i.e., "DIScovering Comparable items with Rules to Explain Treatment Effect", for estimating ITE. In line with its nomenclature, DISCRET specializes in estimating ITE for a specific target sample, leveraging the average treatment effect (ATE) estimated from a subgroup of similar samples obtained through rule-based explanations from the database. Typically, this database comprises all training samples. This process is visually elucidated in Figure 1 using an example from the Infant Health and Development Program (IHDP) dataset (Hill, 2011). As depicted in the figure, when determining the impact of high-quality child care (the treatment variable) on an individual subject, such as a premature infant, the initial step involves extracting a group of comparable premature infants from the database using logical rules. Subsequently, the ITE for the target sample is assessed by applying the

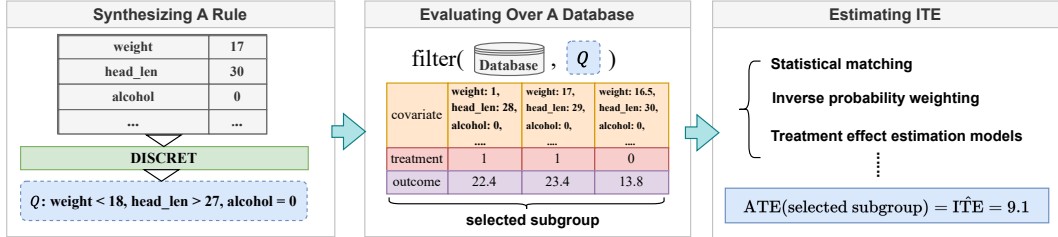

Figure 1: Illustrations of DISCRET using IHDP dataset. To estimate ITE for one sample, DISCRET synthesizes a rule as one explanation, "weight > 1694, head circumference > 27.8, alcohol = 1" which captures a pattern of infants with body weight over 1694 grams, the head circumference over 27.8 centimeters and their mothers taking alcohol during pregnancy. Evaluating this rule over the database retrieves a subgroup of three similar infants. We then estimate the average treatment effect (ATE) over those similar infants as the estimated ITE for the sample shown in the leftmost box.

ATE computed over these similar samples. This approach mirrors the clinical practice of treatment recommendation for individual patients (Seymour et al., 2019), which relies heavily on evaluating treatment effects in patients with analogous characteristics.

To guarantee accurate ITE estimation, we further proposed to leverage Deep reinforcement learning with a novel and tailored reward function for rule learning, which can also overcome the non-differentiability issue caused by the step of evaluating rules over the database. Furthermore, our proposed framework excels in generating inherently more faithful explanations compared to the current state of the art, as quantitatively evaluated through the faithfulness measures introduced by Dasgupta et al. (2022).

We evaluate the capabilities of DISCRET through comprehensive experiments spanning various data domains, including tabular data, text data, and image data, Notably, our approach not only achieves performance levels comparable to black-box models but also excels in producing more faithful explanations than the state of the art. Our contributions can be summarized as follows:

1. We introduce DISCRET, a novel self-interpretable ITE estimation framework that leverages the estimated Average Treatment Effect (ATE) over similar samples retrieved through rule-based explanations for ITE estimation.
2. We present a deep reinforcement learning algorithm tailored for generating explanations specifically designed for ITE estimation.
3. We conduct an extensive series of experiments across diverse data settings, encompassing tabular, text, and image data, providing empirical evidence that DISCRET excels in generating faithful explanations without compromising ITE estimation performance.

## 2 BACKGROUND AND PROBLEM STATEMENT

We first introduce our problem settings. Suppose each sample consists of (i) the pre-treatment covariate variable $X$, (ii) the treatment variable $T$, (iii) a dose variable $S$ associated with $T$, and (iv) observed outcome $Y$ under treatment $T$ and dose $S$. Note that we embrace a versatile framework throughout this study, where $T$ can take on either discrete or continuous values, $S$ is inherently continuous but can be either present or absent, $Y$ can be discrete or continuous, and $X$ may incorporate structured features as well as unstructured features, such as text or image data. In the remainder of the paper, we primarily explore the setting where $Y$ is a continuous variable. A generalization of these settings can be found in the Appendix A.

Given the diversity inherent in treatment and dose variables, the range of treatment effects that need estimation also exhibits significant variability. In this paper, we focus on addressing this diversity by delving into three representative settings:

1. Tabular data with a binary treatment variable $T$ and no dose variables. In this setting, $T = 1$ represents treated unit while $T = 0$ represents untreated unit, and the ITE is defined as the difference of outcomes under the treatment and under the control, respectively (i.e., $\text{ITE}(x) = y_1(x) - y_0(x)$, where $y_1(x)$ and $y_0(x)$ represents the potential outcome with and without receiv-

ing treatment for a sample $x$). The average treatment effect, ATE, is the sample average of ITE across all samples (i.e., ATE $= \mathbb{E}[\text{ITE}]$).

2. Tabular data with a continuous treatment variable $T$. Following Zhang et al. (2022), the average dose-response function is defined as the treatment effect, i.e., $\mathbb{E}[Y|X, do(T = t)]$.

3. Tabular data with a discrete treatment variable $T$ with one additional continuous dose variable $S$. Following Zhang et al. (2022), the average treatment effect is defined as the average dose-response function: $\mathbb{E}[Y|X, do(T = t, S = s)]$.

Furthermore, beyond the treatment effect definitions, the propensity score, represented as the probability of treatment assignment $T$ conditioned on the observed covariates $X$, often plays a pivotal role in regularizing the treatment effect estimation. This propensity score is denoted as $\pi(T|X)$.

Unlike conventional prediction tasks, we are unable to directly observe the counterfactual outcomes during the training phase, rendering the ground-truth treatment effect typically unavailable. To address this challenge and ensure the causal interpretability of our estimated treatment effect, we adhere to the following assumptions proposed by Rubin (1974):

**Assumption 1.** *(Strong Ignorability/Unconfoundedness)* $Y(T = t) \perp T|X$. *In the binary treatment case,* $Y(0), Y(1) \perp T|X$.

**Assumption 2.** *(Positivity/Overlap)* $0 < \pi(T|X) < 1, \forall X, \forall T$.

**Assumption 3.** *(Consistency) For the binary treatment setting,* $Y = TY(1) + (1 - T)Y(0)$.

For the covariate variable $X$, we assume that it is composed of $m$ features, $X_1, X_2, \ldots, X_m$, which could be the categorical or numeric attributes from tabular data or pre-processed features extracted from the text data or image data. We then build logic rule-based explanations upon those features to construct our treatment effect estimator. Those logic rules are assumed to be in the form of $K$ disjunctions of multiple conjunctions, i.e., $R :\text{-} R_1 \vee R_2 \vee \cdots \vee R_H$ where each $R_i$ is a conjunction of $K$ literals: $R_i :\text{-} l_{i1} \wedge l_{i2} \wedge l_{i3} \wedge \cdots \wedge l_{iK}$, and :- means if. In the definition of $R_i$, each $l_{ij}(j = 1, 2, \ldots)$ represents a literal taking the form of $l_{ij} = (A\ op\ c)$, in which $A \in \{X_1, X_2, ..., X_m\}$, and $op$ is equality for categorical attributes while $op \in \{<, >, =\}$ for numeric attributes.

To facilitate ITE estimation, there are some desired properties for the rule $R$:

1. **Local interpretability**: We generate rule-based explanations for each individual sample rather than for a population of samples. Thus, the explanations may differ in different patients. Given a target sample with covariate $x$, we will use $R_x$ to denote the generated rule for this sample. We slightly abuse the notation below by referring to a sample with covariate $x$ as sample $x$.

2. **Satisfiability**: For any rule $R_x$ used for estimating ITE for the sample $x$, this sample's covariates should also satisfy this rule. This guarantees that the sample $x$ and similar samples retrieved by rules share the same characteristics.

3. **Low-bias and Non-emptiness**: We expect that $R_x$ can retrieve a set of similar samples so that the bias between the estimated treatment effect over them and the ground-truth ITE is as small as possible, which is referred to as the **Low-bias** property. In addition, there should be at least one sample from the database whose covariates satisfy these rules, which is the **Non-emptiness** property.

Our objective is to estimate the ITE for a given sample $x$, even in the absence of knowledge regarding its true treatment assignments and outcomes. To achieve this, we aim to develop a rule $R_x$, which adheres to **Local interpretability**, **Satisfiability**, **Low-bias**, and **Non-emptiness**. Once we have identified and established this rule, we proceed by selecting a subgroup of patients who exhibit similarity according to this rule. Subsequently, we employ the estimated Average Treatment Effect (ATE) computed over this subgroup as the estimated ITE for the sample $x$.

## 3 THE FRAMEWORK OF DISCRET

In this section, we provide a step-by-step solution to the rule learning problem as formally introduced in Section 2. Initially, we assume a scenario where each target sample $x$ corresponds to a single disjunction in $R_x$. Thus, our representation of $R_x$ takes the form of $R_x :\text{-} l_1 \wedge l_2 \wedge l_3 \wedge \cdots \wedge l_K$. Within this section, as outlined in Algorithm 1, we commence by elucidating the process of generating $R_x$ and estimating ITE during the inference phase. Subsequently, we delve into the utilization of deep

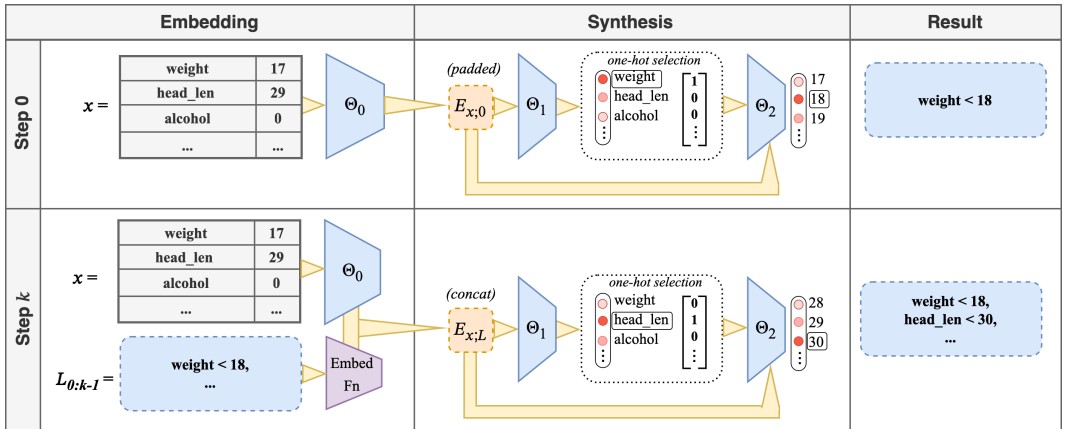

Figure 2: Illustrative overview of the iterative inference forward pass of DISCRET. The model begins at Step 0 by using an embedding of $x$ to generate a single literal For any subsequent step $k$, the model generates rule $L_k$ by embedding all of the literals generated so far, concatenating it to the embedding for $x$, generating the next literal, and appending that literal to the rule $L_{k-1}$.

---

**Algorithm 1** Overview of inference and training phase

---

1: **for** each sample with covariate $x$ **do**
2:     Generate rule $R_x$ for $x$
3:     Retrieve similar patients $R_x(\mathcal{D})$ by evaluating $R_x$ on a database $\mathcal{D}$
4:     **if** Inference **then**
5:         Estimating ATE of $R_x(\mathcal{D})$ as the estimated ITE for $x$
6:     **else if** Training **then**
7:         Collecting rewards based on $R_x(\mathcal{D})$ and performing Deep Reinforcement Learning
8:     **end if**
9: **end for**

---

reinforcement learning techniques to effectively learn $R_x$ while ensuring it adheres to the desired properties outlined in Section 2.

### 3.1 RULE GENERATION AT INFERENCE PHASE

DISCRET generates $R_x$ recursively by learning all of $l_k, k = 1, 2, \cdots, K$ one after the other in multiple rounds. Since $l_k$ takes the form of $(A \; op \; c)$, then at each round $k$, we determine which feature $A$ to select, which constant $c$ to be compared against, and with which operator $op$.

Figure 2 depicts how a literal $l_k$ is generated at the $k_{th}$ round with one running example from IHDP dataset. First, we encode the covariate $x$ as $E_x$ with one neural network parameterized by $\Theta_0$, and also encode all the literals generated in the first $k - 1$ rounds, denoted by "$L_{1:k-1} = \{l_1, l_2, \cdots, l_{k-1}\}$", as $E_L$. Due to space limit, we leave the details of how to encode $L_{1:k-1}$ to the Appendix A. $E_x$ and $E_L$ are then fed into a feed-forward neural network (parameterized by $\Theta_1$) and *the most probable* feature from $\{X_1, X_2, \ldots, X_m\}$ is selected as $A$ for $l_k$.

The selection of the constant $c$ and the operator $op$ for literal $l_k$ depends on the type of the selected attribute $A$. If $A$ is a categorical attribute, then we assign $c$ as $x[A]$, i.e., the value of attribute $A$ in $x$, and assign $op$ as equality, which can guarantee the **Satisfiability** of $R_x$ on $x$.

Instead, if $A$ is a numeric attribute, we first discretize the range of $A$ with a list of $\mu$ evenly distributed float numbers, $\{C_1, C_2, \ldots, C_\mu\}$. We then design a feed-forward neural network (parameterized by $\Theta_2$) to produce the probability distribution of these $\mu$ numbers, which takes the encoding of the covariates, $E_x$, the encoding of $L_{1:k-1}$, $E_L$, and the one-hot encoding of feature $A$ as the model input. *The most probable $C_j$* is then selected as $c$. We further present the extension of the above rule generation process to multiple disjunctions in Appendix A.

Note that instead of outputting the float number $c$ directly, generating the probability distribution of $|C|$ discrete numbers can facilitate the generation of Q value for deep Q learning. After the feature $A$ and the constant $c$ are identified, the operator $op$ is then determined by comparing the value $x[A]$ and $c$. If $x[A]$ is greater than $c$, then $op$ is assigned as $\geq$ and $\leq$ otherwise. This can again guarantee the **Satisfiability** of the rule $R_x$ on $x$.

To handle text and image data, one essential step is to extract features from them prior to the above rule generation process, which then become parts of the covariate variables. For text data, we can extract Bag-of-word (BoW) features; for image data, we may follow concept learning methods (Fel et al., 2023; Tran et al., 2022) to extract concept features.

### 3.2 ITE ESTIMATION AT INFERENCE PHASE

Given a sample $x$ (e.g., a patient) with $(x, t, s, y)$, after generating a rule $R_x$ for a sample $x$, the next step is to evaluate this rule on a database $\mathcal{D}$ to retrieve a subgroup of similar samples, which is denoted by $R_x(\mathcal{D}) = \{(x_i^*, t_i^*, s_i^*, y_i^*)\}_{i=1}^n$. The ITE of the sample $x$ is then estimated with the average treatment effect (ATE) estimated within this subgroup, which is formulated as below for different types of treatment variables and dose variables:

1. With a binary treatment variable and no dose variable, we can estimate the ATE of $R_x(\mathcal{D})$ via arbitrary treatment effect estimation methods in the literature, such as the classical statistical matching algorithm (Kline & Luo, 2022), or the state-of-the-art neural network models. In this paper, we take the K-Nearest Neighbor Matching by default for estimating ATE of $R_x(\mathcal{D})$: ITE $= y_1(x) - y_0(x)$. We can also obtain the estimated outcome by averaging the outcome of samples from $R_x(\mathcal{D})$ with the same treatment as the sample $x$, i.e.:

$$\widehat{y}(t) = \frac{1}{\sum \mathbb{I}(t_i^* = t)} \sum \mathbb{I}(t_i^* = t) \cdot y_i^* \tag{1}$$

2. With a continuous treatment variable $T$ but without dose variables, according to Section 2, the ITE is represented by the outcome conditioned on the observed treatment. One straightforward way to estimate it is to employ the average outcome of samples within $R_x(\mathcal{D})$ that receive similar treatments to $x$, which is also the estimated outcome for this sample:

$$\hat{y} = \frac{\sum \mathbb{I}[(x_i^*, t_i^*, y_i^*) \in \text{top}_k(R_x(\mathcal{D}))] \cdot y_i^*}{\sum \mathbb{I}[(x_i^*, t_i^*, y_i^*) \in \text{top}_k(R_x(\mathcal{D}))]}, \tag{2}$$

in which $\text{top}_k(R_x(\mathcal{D}))$ is constructed by finding the top-$k$ samples from $R_x(\mathcal{D})$ with the most similar treatments to $x$. But again, any existing treatment effect estimation methods for continuous treatment variables from the literature are applicable to estimate $\widehat{\text{ITE}}_x$.

3. With a discrete treatment variable $T$ and one associated continuous dose variable $S$, ITE is estimated in a similar way to equation 2, which is included in Appendix A due to space limit.

Note that one implicit assumption that we made here is that the outcome variable is a continuous variable. The generalization of the above formulas to discrete outcome variables is discussed in the appendix. We can also estimate the propensity score for discrete treatment variables by simply calculating the frequency of every treatment within $R_x(\mathcal{D})$: $\widehat{\pi}(T = t | X = x) = \sum \mathbb{I}(t_i^* = t) / |R_x(\mathcal{D})|$:

### 3.3 TRAINING PHASE

To preserve **low-bias** property, we need to guide the generation of rules such that the estimated ITE is as accurate as possible. We therefore need to optimize this objective by training the three models ($\Theta_0$, $\Theta_1$ and $\Theta_2$) used in the rule generation process. One difficulty of learning $\Theta = [\Theta_0, \Theta_1, \Theta_2]$ is the non-differentiability issue caused by the step of evaluating $R_x$ over the database. We overcome this issue by formulating this model training problem as a deep reinforcement learning (RL) problem, and propose to adapt deep Q learning (DQL) algorithm to solve this problem.

**RL problem formulation**   We first map the notations from Section 3.1 to classical RL terminology. An RL agent takes one *action* at one *state*, collects a *reward* from the environment, which is then transitioned to a new state. In our rule learning setting, a *state* is composed of the covariates $x$ and the generated literals in the first $k - 1$ rounds, $L_{1:k-1}$. With $x$ and $L_{1:k-1}$, the model $\Theta_1$ and $\Theta_2$ collectively determine the $k_{th}$ literal, $l_k$, which is regarded as one *action*. Our goal is then to learn a

policy parameterized by $\Theta$, which models the probability distribution of all possible $l_k$ conditioned on the state $(x, L_{1:k-1})$, such that the value function calculated over all $K$ rounds is maximized:

$$V_{1:K} = \sum\nolimits_{k=1}^{K} r_k \gamma^{k-1}, \tag{3}$$

in which $\gamma$ is a discounting factor. Note that there are only $K$ horizons/rounds in our settings since the number of conjunctions in the generated rules is limited. To bias rule generation towards accurate estimation of ITE, we expect that the value function $V_{1:K}$ reflects how small the ITE estimation error is. However, since the counterfactual outcomes are not observed in the training phase, we therefore use the errors of the observed outcomes as a surrogate of the ITE estimation error. Also, we give a zero reward to the case where the retrieved subgroup, $L_{1:K}(\mathcal{D})$, is empty after evaluating $L_{1:K}$ on the database $\mathcal{D}$, which can thus guarantee the **non-emptiness** of the rules. As a result, $V_{1:K}$ is formulated as

$$V_{1:K} = e^{-\alpha(y - \widehat{y}_{1:K})^2} \cdot \mathbb{I}(|L_{1:K}(\mathcal{D})| > 0), \tag{4}$$

in which $\widehat{y}_{1:K}$ represents the estimated outcome by using the generated rule composed of literals $L_{1:K}$ and $\alpha$ is a hyper-parameter. As a consequence, the reward collected at the $k_{th}$ round of generating $l_k$ becomes $r_k = (V_{1:k} - V_{1:k-1})/\gamma^{k-1}$.

**Deep Q learning for model training**   To maximize the value function $V_{1:K}$, Deep Q learning (DQL) (Mnih et al., 2013) is employed to learn the parameter $\Theta$. To facilitate Q learning, we estimate the Q value with the output logits of the models given a state $(x, L_{1:k-1})$ and an action $l_k$. Further details on how to employ Deep Q learning for model training are included in Appendix A.

We further present some strategies to optimize the design of the cumulative reward function defined in equation 4, which includes incorporating estimated propensity scores into this formula and automatically fine-tuning its hyper-parameters.

**Regularization by estimating propensity scores**   First of all, similar to prior studies on ITE estimation (Shi et al., 2019; Zhang et al., 2022), we regularize the reward function $r_k$ by integrating the estimated propensity score, $\widehat{\pi}(T = t|X = x)$. Specifically, for discrete treatment variables, we reweight equation 5 with the propensity score as a regularized reward function, i.e.:

$$V_{1:K}^{reg} = [e^{-\alpha(y - \widehat{y}_{1:K})^2} + \beta \cdot \widehat{\pi_{1:k}}(T = t|X = x)] \cdot \mathbb{I}(|L_{1:K}(\mathcal{D})| > 0), \tag{5}$$

**Automatic hyper-parameter fine-tuning**   We further studied how to automatically tune the hyper-parameter $\alpha$ and $\beta$ in equation 5. For $\alpha$, at each training epoch, we identify the training sample producing the median of $(y - \widehat{y}_{1:K})^2$ among the whole training set and then ensure that for this sample, equation 4 is 0.5 through adjusting $\alpha$. This can guarantee that for those training samples with the smallest or largest outcome errors, equation 4 approaches 1 or 0 respectively.

We further design an annealing strategy to dynamically adjust $\beta$ by setting it as 1 during the initial training phase to focus more on treatment predictions, and switching it to 0 so that reducing outcome error is prioritized in the subsequent training phase.

## 4 EXPERIMENTS

### 4.1 EXPERIMENTAL SETTINGS

**Datasets**   The experiments are conducted across tabular, NLP, and image settings.

In the context of tabular data settings, we conduct comprehensive evaluations across the settings with diverse categories of treatment variables. Specifically, we have selected the following datasets for our evaluation: the IHDP dataset (Hill, 2011), featuring a binary treatment variable without dose variables; the TCGA dataset (Weinstein et al., 2013), which includes multiple discrete treatment options and continuous dosage information; and two additional datasets, IHDP-C (a variant of IHDP) and News, both containing continuous treatment variables but lacking dose-related information.

For text data, we conducted experiments on the Enriched Equity Evaluation Corpus dataset, abbreviated as EEEC designed to explore the impact of treatment variables, specifically changes in racial

and gender-related nouns, on the predictions of five distinct mood states for individual sentences. Within our experiments, we consider two variants of this dataset, each focusing on racial nouns and gender-related nouns respectively, denoted as EEEC (Race) and EEEC (Gender) respectively.

In our exploration of image data, we utilized a satellite image dataset enriched with associated covariates, originating from various regions of Uganda, succinctly referred to as Uganda (Jerzak et al., 2023b;a). This dataset was used to investigate the potential impact of grant assignments, considered as the treatment variable, on the aggregated summary of skilled labor in the respective regions. The detailed description of these datasets is included in Appendix C.

**Baseline methods**    We consider the following categories of baseline methods. For neural network models, we used TransTEE (Zhang et al., 2022), Dragonnet (Shi et al., 2019), TARNet (Shalit et al., 2017), Ganite (Yoon et al., 2018), DRNet (Schwab et al., 2020), and VCNet (Nie et al., 2020). Note that Ganite and VCNet can only handle binary treatments without dose variables, which are thus not applied to TCGA, IHDP-C, and News dataset.

We also consider self-interpretable models such as logistic regression (LR), decision tree (DT), and random forests (RF) model, which are integrated into R-learner (Nie & Wager, 2021) for causal inference. For tree-based models, we limit the complexity (e.g., the number of trees and tree depths) to be the same as DISCRET for fair comparison. In addition, we adapted three self-interpretable models designed for general prediction tasks to ITE estimations. These three models include ENRL (Shi et al., 2022), ProtoVAE (Gautam et al., 2022)[1] and Neural Additive Model (NAM) (Agarwal et al., 2021), which generate rules, prototypes and feature attributes as explanations respectively.

We further compare DISCRET against post-hoc explainers, including Lore (Guidotti et al., 2018), Anchor (Ribeiro et al., 2018), Lime (Ribeiro et al., 2016), Shapley values (Shrikumar et al., 2017) and decision tree-based model distillation methods (Frosst & Hinton, 2017) (Model Distillation for short). In the experiments, we apply these methods to the TransTEE model during test time.

**Evaluation metrics**    The first type of metrics is for evaluating ITE estimation performance. For the datasets with binary treatment variables, by following prior studies (Shi et al., 2019; Shalit et al., 2017), the absolute error in average treatment effect, i.e.,: $\epsilon_{ATE} = |\frac{1}{n} \sum_{i=1}^{n} ITE(x_i) - \frac{1}{n} \sum_{i=1}^{n} \widehat{ITE}(x_i)|$, is employed for evaluations. Both in-sample and out-of-sample $\epsilon_{ATE}$ are reported. For the datasets with either continuous dose variables or continuous treatment variables, we follow (Zhang et al., 2022) to report the average mean square errors $AMSE$ between the ground-truth outcome and predicted outcome on the test set. The definition of $AMSE$ is provided in Appendix F. For the image dataset, Uganda, since there is no ground-truth ITE, we therefore only report the average outcome errors between the ground-truth outcomes and the predicted outcomes conditioned on observed treatments, i.e., $\epsilon_{\text{outcome}} = \frac{1}{n} \sum_{i=1}^{n} |y_i - \hat{y}_i|$.

We further compare the *faithfulness* of the explanations produced by DISCRET, the post-hoc explainers, and self-interpretable models. Specifically, we leverage two faithfulness metrics, *consistency* and *sufficiency*, proposed by (Dasgupta et al., 2022). Roughly speaking, *Consistency* quantifies how consistent the model predictions are between samples with the same explanations while *sufficiency* generalizes this notion of consistency to arbitrary samples satisfying the same explanations (but not necessarily producing the same explanations). The formal definition of these two metrics and how to evaluate them are discussed in Appendix F.

**Configurations for DISCRET**    We perform grid search on the number of conjunctions, $K$, and the number of disjunctions, $H$. We incorporate the propensity score regularization and auto-tuning on $\alpha$ and $\beta$. But we also performed an ablation study in Appendix D to study how these optimizations on the reward functions influence the ITE estimation performance.

**Extracting features from text and image data**    For text data, we employ the word frequency features such as "Term Frequency-Inverse Document Frequency" (Baeza-Yates et al., 1999). For image data, we follow (Fel et al., 2023) to extract interpretable concepts as the features, which is discussed in Appendix E in detail.

---

[1]note that ProtoVAE is designed for image data. We therefore only compare DISCRET against this method in Uganda dataset

| Modality → | Tabular | | | | | Text | Image |
|---|---|---|---|---|---|---|---|
| Dataset → | IHDP | | TCGA | | IHDP-C | EEEC (Race) | Uganda |
| Method ↓ | $\epsilon_{\text{ATE}}$ (In-sample) | $\epsilon_{\text{ATE}}$ (Out-of-sample) | $\epsilon_{\text{ATE}}$ (In-sample) | $\epsilon_{\text{ATE}}$ (Out-of-sample) | AMSE | $\epsilon_{\text{ATE}}$ | $\epsilon_{\text{outcome}}$ |
| LR | 3.366±2.189 | 2.497±1.814 | 31.737±0.001 | 57.541±0.001 | 36.640±16.455 | 0.014±0.016 | 1.796±0.021 |
| DT | 0.345±0.273 | 0.530±0.399 | 0.200±0.012 | 0.202±0.012 | 22.136±1.741 | 0.014±0.016 | 1.796±0.021 |
| RF | 0.739±0.284 | 0.737±0.383 | 0.263±0.057 | 0.264±0.058 | 21.348±1.222 | 0.525±0.573 | 1.820±0.013 |
| NAM | 0.225±0.221 | 0.519±0.512 | 4.201±0.232 | 4.211±0.152 | 24.706±0.756 | 0.152±0.041 | 1.710±0.098 |
| ENRL | 4.160±1.060 | 4.439±1.587 | 10.938±2.019 | 10.942±2.019 | 24.720±0.985 | - | 1.800±0.143 |
| Dragonnet | 0.177±0.139 | 0.219±0.143 | - | - | - | 0.011±0.018 | 1.709±0.127 |
| TARNet | 0.186±0.130 | 0.408±0.418 | 1.421±0.078 | 1.421±0.078 | 12.967±1.781 | 0.009±0.018 | 1.743±0.135 |
| Ganite | 1.127±0.481 | 1.144±0.352 | - | - | - | 1.998±0.016 | 1.766±0.024 |
| DRNet | 0.188±0.132 | 0.407±0.422 | 1.374±0.086 | 1.374±0.085 | 11.071±0.994 | 0.008±0.018 | 1.748±0.127 |
| VCNet | 4.205±0.569 | 4.434±0.851 | 0.292±0.074 | 0.292±0.074 | - | 0.011±0.017 | 1.890±0.110 |
| TransTEE | **0.128±0.103** | **0.203±0.130** | **0.055±0.014** | **0.056±0.013** | **0.488±0.288** | 0.003±0.017 | 1.707±0.158 |
| DISCRET | 0.274±0.253 | 0.344±0.303 | 0.076±0.019 | 0.077±0.020 | 0.801±0.165 | **0.000±0.017** | **1.662±0.136** |

Table 1: ITE estimation errors (lower is better). We **bold** the smallest estimation error for each dataset and underline the smallest among the self-interpretable methods. DISCRET outperforms existing self-interpretable methods on 6 out of the 7 benchmarks.

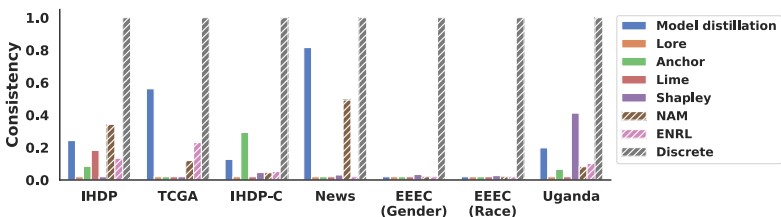

Figure 3: The comparison of the consistency scores between different explanation methods across different datasets. Larger is better.

## 4.2 QUANTITATIVE ITE ESTIMATION PERFORMANCE

We include the ITE estimation results for tabular setting, NLP setting, and image setting in Table 1. In this table, we bold and underline those performance numbers that are the smallest prediction errors (or closest to the ground-truth ATE) among all the methods and all the self-interpretable methods respectively. Due to the space limit, the results on News dataset and EEEC (Gender) dataset are not reported in Table 1, which are included in Table 3 and 5 respectively in Appendix G.

As Table 1 shows, DISCRET outperforms all the self-interpretable methods most times, thus demonstrating the performance advantage of DISCRET over these methods. By comparing against black-box models, DISCRET only performs slightly worse than black-box models in most cases. DISCRET can even outperform them on Uganda dataset, which is possibly caused by the overfitting issue encountered by those black-box models due to the small size of Uganda.

## 4.3 QUANTITATIVE EXPLANATION QUALITY EVALUATIONS

We evaluate the *consistency* and *sufficiency* on the explanations produced by DISCRET, the state-of-the-art self-interpretable models, and the post-hoc explainers. For those explainers producing feature-based explanations, we also follow (Dasgupta et al., 2022) to discretize the feature importance scores, say, by selecting the Top-K most important features, for identifying samples with exactly the same explanations. For fair comparison, we evaluate the explanations generated w.r.t. the same set of features extracted from NLP and image data.

Due to the space limit, we only report the consistency scores of different explanation methods across all the settings, in Figure 3. The full consistency scores are included in Table 4 in Appendix G. As this figure indicates, DISCRET always achieves 100% consistency since the same explanations in DISCRET always retrieve the same subgroup from the database, thus generating the same model predictions. In contrast, the baseline explanation methods generally have extremely low consistency scores in most cases. We also include the sufficiency score results in Table 7, which shows that DISCRET can still obtain higher sufficiency scores in most cases than other explanation methods.

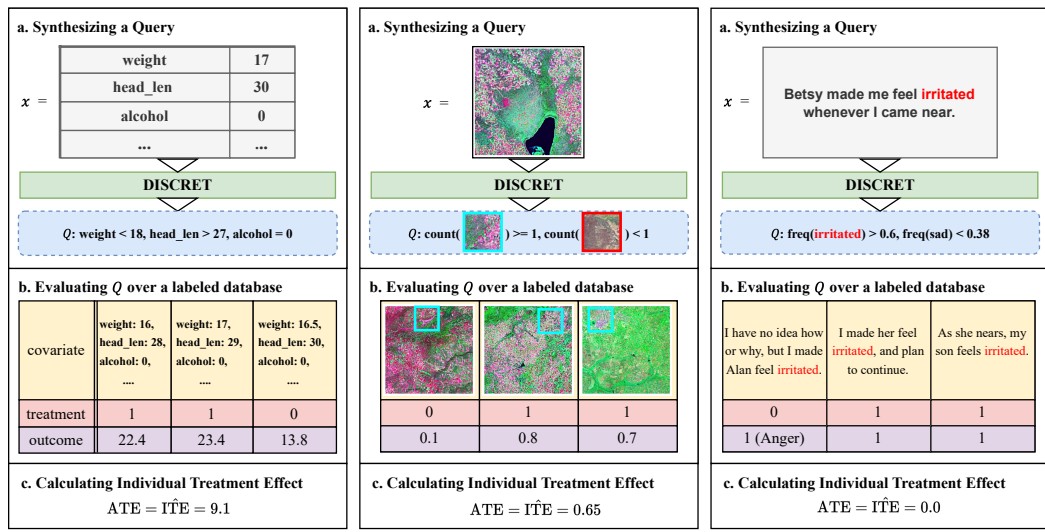

Figure 4: Qualitative examples from IHDP, EEEC and Uganda dataset.

## 4.4 QUALITATIVE ANALYSIS

We perform qualitative studies on some representative examples from tabular setting (IHDP dataset), NLP setting (EEEC dataset), and image setting (Uganda dataset) (shown in Figure 4) to explore whether the learned rules are reasonable or not. For the example from EEEC dataset, "Betsy made me feel irritated whenever I came near" shown in Figure 4, DISCRET generated the following rule constructed over the word frequency features, "freq(irritated) > 0.06, freq(sad) < 0.38". This indicates that sentences following this rule should have non-zero occurrence of the word "irritated" and not too many occurrences of "sad". The evaluation of this rule thus returns three sentences from the database, which all contain the word "irritated". Since the word "irritate" is highly correlated with the mood "Anger", the change of the "Gender" or "Race" nouns won't actually affect the mood of the whole sentence, which matches the ground-truth labels. As a result, the estimated ATE over the subgroup and thus the estimated ITE for this example sentence is 0. For the example from Uganda dataset shown in Figure 4, we leave the analysis to Appendix H due to space limit.

## 5 RELATED WORK

**Treatment effect estimation** A substantial body of research has been dedicated to the estimation of treatment effects through machine learning. For instance, Shalit et al. (2017) introduced a novel theoretical analysis and a comprehensive family of algorithms to predict ITE from observational data. Additionally, Wager & Athey (2018) pioneered the introduction of a non-parametric causal forest approach tailored for the estimation of heterogeneous treatment effects. Moreover, bridging the gap between the predictive power of machine learning models and the need for interpretable decisions remains a pivotal challenge. To address the issue, Kim & Bastani (2019) proposed a framework for learning interpretable models from observational data for predicting ITE.

Due to space limit, we further discuss related work on **Model interpretability** in Appendix B.

## 6 CONCLUSION

In this work, we introduce a self-interpretable framework, DISCRET, for ITE estimation. Within the DISCRET framework, we not only focus on accurate ITE estimation but also prioritize the generation of faithful explanations. To achieve this, we have developed a specialized deep reinforcement learning algorithm that is tailored to the task of generating these explanations. Extensive experiments across different data modalities demonstrate that DISCRET can balance the ITE estimation performance and the failthfulness in explanations.

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
