# OpenReview forum: "DISCRET: a self-interpretable framework for treatment effect estimation"
_ICLR.cc/2024/Conference — Submitted to ICLR 2024_

### Official Review · Reviewer_QYCJ · 2023-10-28

**Soundness:** 2 fair
**Presentation:** 2 fair
**Contribution:** 2 fair
**Rating:** 3
**Confidence:** 4

**Summary:**

The authors first point out that the majority of ITE estimation algorithms use non-interpretable black-box models which does not allow for interpretability. To address this issue, they propose DISCRET, which is a matching type algorithm for ITE estimation. They conduct extensive experiments over tabular, natural language, and image settings, and show empirically, that DISCRET can achieve on par performance as black-box models while generating more faithful explanations.

**Strengths:**

- The paper attempts to address a very important task, i.e., designing interpretable ITE estimation algorithms.
- Formulation of the problem as providing logic rule-based explanations is interesting.
- The idea of using reinforcement learning is interesting.
- The experiments are extensive and cover a wide range of scenarios.

**Weaknesses:**

- Writing of the paper could improve, especially in terms of motivating various design choices involved in developing the algorithm.
- It is not clear how RL is helpful to solve the problem, as I don’t see the need for temporality in evaluating the literals. Isn’t a supervised learning method actually a better fit here?
- I’m not sure if ICLR is the right venue for publishing this paper, since the proposed algorithm has no “representation learning” component.

**Questions:**

- Why train 3 models in the rule-generation process? How is this hyperparameter selected? What is the responsibility of each of them?
- Why/how is there temporality in K literals, to motivate use of RL? It is not clear why one literal should strictly come before another.
- How does the RL framework adapted to help with interpretability?

---

### Official Review · Reviewer_JeTi · 2023-10-31

**Soundness:** 2 fair
**Presentation:** 3 good
**Contribution:** 2 fair
**Rating:** 3
**Confidence:** 5

**Summary:**

The paper introduces DISCRET, a self-interpretable framework designed to estimate individual treatment effects. DISCRET generates rules to identify similar samples to a target and leverages the average treatment effect of these identified samples as the estimated individual treatment effect for the target sample.

**Strengths:**

- The model leverages interpretable rules to estimate individual treatment effects, making it easier to understand model predictions.

- The model demonstrates its effectiveness across diverse domains, including tabular data, natural language processing, and image analysis.

- The code is available for review.

**Weaknesses:**

- The proposed method generates a rule for each sample to provide local interpretability. However, if rules are not mutually exclusive, the same sample may satisfy different rules with varying ITE estimations. Each rule doesn’t fully explain the outcome.

- An analysis of the number of rules hasn’t been conducted. The choice of the number of rules is critical in this approach and highly impacts the estimation performance. For example, when there are few rules, it might lead to an abundance of similar samples, resulting in overgeneralized treatment effect estimation rather than ITE that reflects the unique characteristics of the target sample. On the other hand, if there are many rules, it can provide more accurate ITE estimates, but it may be challenging to differentiate with a matching method.

- The proposed method underperforms compared to ITE models. This suggests that the model may struggle to capture the unique features of data, possibly due to overgeneralization by similar samples. Additionally, is achieving 100% consistency an advantage of the model? This also indicates the model's overgeneralization.

- How to deal with the violation of positivity assumption? The identified subgroup can include patients only from a single treatment group.

- The computational efficiency of the proposed method is a concern. High-dimensional data requires significant computational resources, which is a notable drawback

- Some details of experiments are missing. For example, what are in-sample and out-of-sample?

**Questions:**

See the above section.

---

### Official Review · Reviewer_BpHQ · 2023-11-01

**Soundness:** 3 good
**Presentation:** 3 good
**Contribution:** 3 good
**Rating:** 5
**Confidence:** 5

**Summary:**

This paper presents an interpretable approach for ITE. It is mainly to identify certain clusters of samples which would share similar treatment and control outcomes, and then estimate the ATE from this cluster. Overal I think this approach combines the advantages of reinforcement learning to identify the best clustering prototype, meanwhile the rule based filtering is interpretable and intuitive.

**Strengths:**

1: Although stratification-based ITE has been discussed for many years, it is an innovative idea to build such rules to stratify samples to get clarify on how ML models work.
2: The qualitative study and visualizations in Fig 4 is very interesting.

**Weaknesses:**

1: This model is developed based on IHDP which has treatment and control groups in a similar level. In real practice, how to make sure you always have sufficient sample size in the subgroup of ATE calculation? This subgroup might only have T=1 or T=0 or null if the rule filters out most/all of the samples.
2: There are specific causal inference tree-based models. For decision tree, you should replace it with BART. For RF, you should replace it with causal forest. For LR, there are also variants. Do you use one model for each group, or a single model for both groups?
3: There exists pretty significant performance gaps between the proposed method and the SOTA.
4: Compared to other ICLR submissions I have reviewed, I would suggest the authors to expand on the idea to make the paper more extensive. In my opinion, the experiments and methods lack sufficient depths.

**Questions:**

In addition to the weakness above, I would suggest the authors to delve deeper into the other clustering methods that identifies similar samples to estimate ITE. There are many VAE-based models in this direction, for example https://arxiv.org/abs/2307.03315. Except being more interpretable (as rule-based methods are more intuitive than latent encoders), are there any other improvements/benefits or constraints when using rule-based models?

---

### Official Review · Reviewer_XqUu · 2023-11-01

**Soundness:** 2 fair
**Presentation:** 3 good
**Contribution:** 2 fair
**Rating:** 5
**Confidence:** 3

**Summary:**

The authors propose DISCRET, an interpretable framework for individual treatment effect estimation (ITE) via reinforcement learning. Different from previous works finding rules by segmenting populations, DISCRET directly generates explainable rules for each individual. The generated rules are then used to identify similar samples across treatment and control groups for estimating ITE. In the experiments, the authors show that DISCRET can achieve comparable performance with state-of-the-art black-box model and can also generate rules for tabular, image and text data.

**Strengths:**

1. The idea of individual instead of population based rule generation is interesting. According to the experiments, the generated rules are indeed effective and explainable. Also, the reinforcement learning based solution seems reasonable and elegant.
2. The proposed methods can work on complicated data such as images and text and generate interesting rules.
3. The paper is overall well-written and easy to follow. The presentation of the proposed approach is excellent. The discussions in main papers and appendix are detailed and informative.

**Weaknesses:**

1. It is unclear why the proposed method can significantly outperform decision tree-based methods. It may require further investigation to identify the factors (e.g., better rules) of the performance gap.
2. There are some hyperparameters seem to be important but are not carefully studied in this paper. For example, the distance metric and the value K in K-nearest neighbors.
3. The training cost is unclear, which could be an important metric in a work concerning practical usage.
4. Minor typo: in figure 1, the numbers (e.g., 1694) in the caption and in the leftmost box do not add up.

**Questions:**

1. Following weakness 1, what would be the reason that DISCRET can outperform decision-tree based methods? If the reason is that DISCRET generates more effective rules, I am wondering why the decision-tree based methods fail to find those rules?
2. When training DISCRET, do we need to fix the number of rules and literals in advance?
3. Although non-emptiness is encouraged by the desgined reward function, is it 100% guaranteed that emptiness does not happen after training?
4. If we retrain DISCRET, will the generated rule set be mostly similar?

---

### Meta-Review · Area_Chair_prRX · 2023-12-10

**Metareview:**

While the reviewers appreciated the paper’s  presentation, and initial experiments, their main concerns were with (a) inadequate baselines, (b) lack of identification guarantees for generated rules, (c) lack of ablation studies, (d) missing discussion on computational efficiency, and (e) missing sensitivity analyses. There was no author response. For these reasons I vote to reject. The reviewers have given extremely detailed feedback and I recommend the authors follow / respond to their comments closely before submitting to a future ML venue. If the authors are able to fix these things it will make a much stronger submission.

**Justification For Why Not Higher Score:**

All reviewers argued reject and there was no author response, should have been withdrawn.

**Justification For Why Not Lower Score:**

N/A

---

### Decision · Program_Chairs · 2024-01-16

Reject